# An In-House ELISA for Anti-Porcine Circovirus Type 2d (PCV2d) IgG: Analytical Validation and Serological Correlation

**DOI:** 10.3390/vaccines13060657

**Published:** 2025-06-19

**Authors:** Gyeong-Seo Park, Byoung Joo Seo, Woo Ju Kwon, Yeong Lee Seok, Hyo Jeong Lee, Sung Ho Lee, Minju Kim, MinChul Lee, Chanhee Chae, Chonghan Kim

**Affiliations:** 1Vaccine Lab, WOOGENE Co., Ltd., 775 Gyeongin-Ro, Seoul 07299, Republic of Korea; gyeoungseopark@gmail.com (G.-S.P.); byoungjooseo@gmail.com (B.J.S.); cosmos0021@gmail.com (W.J.K.); selly0202@woogenebng.com (Y.L.S.); lhyoj123@gmail.com (H.J.L.); 2ChoongAng Lab, WOOGENE Co., Ltd., 230 Jeongmunsongsan-Ro, Hwaseong 18628, Republic of Korea; dasal235@gmail.com (S.H.L.); mjk@woogenebng.com (M.K.); lmc7684@woogenebng.com (M.L.); 3College of Veterinary Medicine, Seoul National University, Seoul 08826, Republic of Korea; swine@snu.ac.kr

**Keywords:** porcine circovirus type 2d, in-house antibody production, serological evaluation, in-house ELISA, validation

## Abstract

**Background/Objectives**: Porcine circovirus type 2d (PCV2d) is the predominant genotype associated with porcine circovirus-associated disease (PCVAD), leading to significant economic losses. In South Korea, current vaccine lot-release testing relies on a T/C-ratio-based guinea pig assay, which lacks scientific justification and methodological robustness. This study aimed to develop and validate a statistically defined in-house ELISA using rabbit-derived polyclonal antibodies against PCV2d for the standardized evaluation of immunogenicity. **Methods**: Polyclonal IgG was generated by immunizing a rabbit with inactivated PCV2d, and it was purified through Protein A chromatography. Guinea pigs (*n* = 18) were immunized with IMMUNIS^®^ DMVac, an inactivated PCV2d vaccine candidate developed by WOOGENE B&G, at different doses. In-house ELISA parameters were optimized (antigen coating, blocking agent, and substrate incubation), and analytical performance was evaluated by ROC, linearity, reproducibility, and specificity. Sera from guinea pigs and pigs were analyzed under validated conditions. **Results**: The optimal performance was achieved using 10^5^ genomic copies/mL of the antigen coating and a 5% BSA blocking agent. The assay showed strong diagnostic accuracy (AUC = 0.97), reproducibility (CVs < 5%), and linearity (R^2^ = 0.9890). Specificity tests with PCV2a, PCV2b, and PRRSV showed minimal cross-reactivity (<7%). The cross-species comparison revealed a positive correlation (R^2^ = 0.1815) and acceptable agreement (bias = −0.21) between guinea pig and porcine sera. The validated cut-off (S/P = 0.4) enabled accurate classification across both species and aligned well with commercial kits. **Conclusions:** The in-house ELISA offers a robust, reproducible, and scientifically validated platform for immunogenicity verification, supporting its application in Korea’s national lot-release system. Homologous competition assays with PCV2d are planned to further confirm antigen specificity.

## 1. Introduction

Porcine circovirus (PCV) is a non-enveloped, circular, single-stranded DNA virus belonging to the genus *Circovirus* within the family *Circoviridae* [1]. PCV has been found to have a similar genome structure regardless of the subtype or species. It has two oppositely oriented open reading frames (ORFs) that construct a circular genome. ORF1 (the rep gene) contains genes related to viral replication, while ORF2 (the cap gene) includes genes related to capsid proteins [2]. However, it has also been shown that a similar genomic structure does not mean that the properties of each ORF are identical. Taxonomically, PCVs are classified into four subtypes: *Porcine circovirus* 1 (PCV1), *Porcine circovirus* 2 (PCV2), *Porcine circovirus* 3 (PCV3), and *Porcine circovirus* 4 (PCV4) [3,4]. PCV1 is known to be a cell-culture-derived, non-pathogenic virus [5] and is considered irrelevant to disease [6,7]. In contrast, PCV2 causes PCV2-associated disease (PCVAD) [8,9], porcine respiratory disease complex (PRDC) [10], and post-weaning multisystemic wasting syndrome (PMWS) in swine [11,12]. PCV3, which was first identified in the United States in 2015 through metagenomic sequencing, has been detected with various clinical signs, including porcine dermatitis and nephropathy syndrome (PDNS), systemic inflammatory responses, reproductive disorders, and subclinical infections [4]. PCV4, which was initially reported in China in 2019, has been detected in several Asian countries [13,14,15,16]. Recently, PCV4-infected pigs displayed PDNS, including respiratory and enteric signs [13]; however, the pathogenic significance of PCV4 remains under investigation [17,18].

From a taxonomic perspective, PCV2 has been classified into eight genotypes; PCV2a, PCV2b, PCV2c, PCV2d, PCV2e, PCV2f, PCV2g, and PCV2h [19,20,21]. In South Korea, three genotypes of porcine circovirus type 2 (PCV2), namely PCV2a, PCV2b, and PCV2d, are known to circulate [22,23]. Although PCV2d has largely replaced PCV2a and PCV2b as the predominant circulating genotype, several studies suggest that PCV2a-based commercial vaccines can still confer partial or sufficient protection against PCV2d infection, despite its genetic divergence [24]. PCV2d has emerged as a significant threat to swine health worldwide, with reports of increased virulence and substantial economic losses [25,26]. Surveillance data from farm and slaughterhouse specimens confirm that PCV2d is now the dominant strain in South Korea [23,27]. Moreover, neutralization assays have demonstrated that certain field strains of PCV2d exhibit reduced neutralizing reactivity to antibodies induced by PCV2a-based vaccines, indicating potential antigenic variability within the PCV2d genotype [28]. To mitigate the impact of PCV2d, various vaccination programs—including both PCV2a-based and genotype-adapted formulations—have been actively implemented. However, these efforts have not always yielded consistent immunological outcomes across different field conditions [29,30,31].

Currently, various PCV2 vaccines are commercially available in South Korea [32]. Multinational corporations and domestic veterinary pharmaceutical companies actively develop or commercialize these vaccines. Most of the commercially available vaccines are based on traditional vaccine platforms: inactivated vaccines [33,34]. However, there has recently been growing interest in utilizing advanced platforms, including virus-like particle (VLP) and subunit vaccines, for PCV2 vaccine development [35,36,37]. Once product development is completed, the product undergoes approval procedures by South Korea’s Animal and Plant Quarantine Agency (APQA) to obtain a product license. However, there are several issues with the current permission criteria in South Korea. One of the main issues is the antiserum titer test for PCV2-vaccinated guinea pigs used in the APQA national veterinary drug standard. The results of the test rely on the T/C ratio, where the optical density (OD) value of the vaccinated group (T) is divided by the average OD value of the control group (C). This criterion lacks a scientific basis and might not be appropriate for different pathogens, making it challenging to assess and validate vaccine efficacy and perform standardization procedures accurately.

Therefore, this study aimed to develop and validate an in-house indirect ELISA using rabbit-derived anti-PCV2d IgG, not as a commercial diagnostic kit, but as a scientifically standardized assay platform tailored for internal immunogenicity verification during vaccine production. Through systematic optimization, standardization, and analytical validation, we established an objective and reproducible methodology that aligns with the requirements of Korea’s national lot-release guidelines, offering a statistically grounded alternative to the current T/C-based evaluation system mandated by the Animal and Plant Quarantine Agency (APQA).

## 2. Materials and Methods

### 2.1. Vaccine Preparation Process

A bivalent vaccine, IMMUNIS^®^ DMVac (DMVac, WOOGENE B&G Co., Ltd, Seoul, Republic of Korea), was developed using the inactivated whole porcine circovirus 2d strain (MW623084 SNUVR201901, 1 × 10^5^ FAID_50_/mL) and *Mycoplasma hyopneumoniae* (WGB-Mhp bacterin, O.D. 0.12 at 410 nm). The FAID_50_ (50% fluorescence active infectious dose) refers to the viral titer determined by fluorescence-based detection of infected cells, following a standard method previously described in the context of SFTSV (severe fever thrombocytopenia syndrome virus) quantification [38]. Although the 50% fluorescence antibody infectious dose (FAID_50_) is not directly equivalent to the 50% tissue culture infectious dose (TCID_50_) or fluorescent focus-forming units (FFUs), it serves as a semi-quantitative indicator of infectious viral concentration, based on the cytopathic effect detected by immunofluorescence staining. The bivalent vaccine was adjuvanted with Carbopol (2.0mg/mL, Lubrizol, Wickliffe, OH, USA) and Saponin (2.0 µg/mL, Sigma-Aldrich, St Louis, MO, USA) as previously described [33,39]. Porcine circovirus 2d was propagated in the porcine kidney cell line PK-15 that was free from PCV1 contamination using the MEM-α medium (Gibco^®^ MEM-α, Life Technologies, Carlsbad, CA, USA) supplemented with 5% heat-inactivated fetal bovine serum (FBS, Life Technologies, USA), an antibiotic–antimycotic cocktail (Anti-Anti, Life Technologies, USA), and 0.25 µg/mL Fungizone^®^ [amphotericin B] at 37 °C in a humidified 5% CO_2_ environment. This culture medium is referred to as the MEM-α growth medium. Viral titration was conducted on PCV1-free PK-15 cells, and the titers were determined at 4 to 5 days post-inoculation based on the fluorescent antibody technique with the focus-forming units (FFUs) as previously described [40,41]. The PCV2d-virus-containing cell lysate was clarified by means of centrifugation at 5000 rpm for 30 min at 4 °C. The clarified samples were filtered through a filter with a pore size of 0.22 µm and collected in a sterile container.

### 2.2. Quantitative Real-Time Polymerase Chain Reaction of PCV2d DNA for Viral Titration

A clarified virus was extracted using the Viral-Gene Spin^TM^ Viral DNA/RNA Extraction Kit (iNtRON Biotechnology, Seongnam, Republic of Korea) according to the manufacturer’s instructions. Viral nucleic acids were extracted and analyzed using the Prime-Q PCV2/PRRSV Detection Kit (GenetBio, Daejeon, Republic of Korea), which employs a multiplex RT-qPCR protocol for the simultaneous detection of PCV2 (a DNA virus) and PRRSV (an RNA virus).

### 2.3. Phylogenetic Analysis

The *ORF2* region of the extracted PCV2d genome was amplified using sequencing primers as previously described [23]. The PCR products were then sequenced using a commercialized sequencing service (COSMO GENETEC, Seoul, Republic of Korea). The full-length PCV2d *ORF2* gene was assembled and quality-checked using SeqMan Pro (Lasergene, DNASTAR, Madison, WI, USA) to ensure the accurate alignment and error correction of the sequencing data. A phylogenetic tree for PCV2 genotypes, including reference strains, was constructed using the maximum likelihood (ML) methods with 1000 replicates for the bootstrap value based on the *ORF2* gene and utilizing the software MEGA 11.

### 2.4. Animal Study

The animal experiments were performed on 1 rabbit and 18 guinea pigs (Figure 1A,B). The animal experiment protocol was approved by the WOOGENE B&G Co., Ltd. Institutional Animal Care and Use Committee (WG-IACUC-2024-004) and performed following the guidelines and regulations detailed by the committee.

#### 2.4.1. Production of In-House Polyclonal Antibodies (Rabbit Anti-PCV2d IgG)

One 3-month-old female NZW (New Zealand White) rabbit was purchased from Samtako Co., Ltd. (Osan, Republic of Korea) to produce in-house antibodies. After a 3-day acclimatization period, the rabbit was subcutaneously (S.C.) inoculated with inactivated 10^7^ FAID_50_/mL of the PCV2d (SNUVR201901) whole virus emulsified with Complete Freund’s Adjuvant (CFA, vac-cfa-10, InvivoGen, San Diego, CA, USA) and Incomplete Freund’s Adjuvant (IFA, vac-ifa-10, InvivoGen, San Diego, CA, USA). The use of Complete Freund’s Adjuvant (for primary immunization) and Incomplete Freund’s Adjuvant (for boosters) was approved exclusively for the rabbit study by the Institutional Animal Care and Use Committee of WOOGENE B&G Co., Ltd. (Approval No. WG-IACUC-2024-004). The procedure was classified as a Category D experiment, and xylazine was administered to ensure appropriate sedation and pain control. The inoculation was administered three times at 2-week intervals. The rabbit’s condition was recorded until the end of the experiment. Euthanasia was performed via high concentrations of carbon dioxide (CO_2_) following AVMA (American Veterinary Medical Association) and NRC (National Research Council) guidelines [42]. After euthanasia, blood was collected and used for an in-house antibody purification process.

#### 2.4.2. Evaluation of PCV2d-Specific Antibodies by Guinea Pig Antiserum

To evaluate the PCV2d-specific antiserum due to vaccination, eighteen 5-week-old female Hartley guinea pigs were purchased from Samtako Co., Ltd. (Republic of Korea), and randomly divided into 6 groups (Figure 1B). After 3 days of acclimatization, Group A (0.1 ml, 1/10 dose of inactivated 10^6^ FAID_50_/mL of PCV2d), Group B (0.1 mL, 1/10 dose of inactivated 10^5^ FAID_50_/mL of PCV2d [IMMUNIS^®^ DMVac]), Group C (0.5 mL, 1/4 dose of inactivated 10^5^ FAID_50_/mL of PCV2d [IMMUNIS^®^ DMVac]), Group D (1.0 mL, 1/2 dose of inactivated 10^5^ FAID_50_/mL of PCV2d [IMMUNIS^®^ DMVac]), and Group E (2.0 mL, 1 dose of inactivated 10^5^ FAID_50_/mL of PCV2d [IMMUNIS^®^ DMVac]) were intramuscularly (I.M) inoculated with the DMVac at 0 and 2 weeks post-vaccination (WPV). Group F (control) was administered PBS only, without the adjuvant. The condition of the guinea pigs was recorded until the end of the experiment. Euthanasia was performed via high concentration of carbon dioxide (CO_2_) following the AVMS (American Veterinary Medical Association) and NRC (National Research Council) guidelines [42]. After euthanasia, blood was collected on the designated days to determine the guinea pig antiserum titer against PCV2d.

#### 2.4.3. Application of Guinea Pig and Field Pig Sera for ELISA Validation

Guinea pigs (n = 3/group) were immunized intramuscularly with 1/4, 1/2, and 1 doses of an PCV2d vaccine (10^5^ FAID_50_/mL). Serum samples collected at 0, 7, 14, and 21 days post-vaccination (DPV) were tested using the in-house and commercial kits. S/P ratios were compared longitudinally, and differences were assessed by two-way repeated-measures ANOVA.

To evaluate diagnostic agreement, field pig sera (n = 10 per year from 2022, 2023 and 2024) provided by the College of Veterinary Medicine, Seoul National University, and previously classified by a commercial kit were tested using the in-house ELISA. Results were grouped as positive, false positive, or negative and compared with commercial kit classifications. The use of archived sera was approved by the institutional committee, and all samples were provided with prior consent for research purposes under institutional guidelines (Seoul National University).

### 2.5. Purification and Optimization of an In-House Antibody (WG-PCV2d pAb)

Antisera (200 mL) obtained from rabbits immunized with PCV2d were subjected to an in-house antibody purification process using the AKTA pure™ M 150 system equipped with a fraction collector (F9-R, Cytiva, Marlborough, MA, USA) following the manufacturer’s guidelines [41]. The purification process was conducted using a Protein A affinity column (HiTrap Protein A HP, Cytiva, USA) with a binding capacity of approximately 20 mg/mL, along with the prepared binding/washing buffer (Buffer A, 20 mM sodium phosphate at pH 7.0), elution buffer (Buffer B, 0.1 M citric acid at pH 3.0), and equilibrium buffer (Buffer C, 1 M Tris-HCl at pH 9.0). The prepared solutions were filtered through a 0.45 µm filter before being used in the purification process. The Protein A affinity step was performed using the UNICORN software v 7.3 (Cytiva, Marlborough, MA, USA), and the process was carried out in the following sequence: column washing, sample application, elution, and equilibration. All steps were monitored in real time through the system control of the UNICORN software, which tracked ultraviolet (UV) absorbance at 280 nm (A280), conductivity (Cond), fraction, injection, pH, and system pressure. During the elution step, all fractions were collected using a fraction collector. After observing the elution peak on the software, Buffer C was added, and the collected fractions were stored in a single sterilized tube.

### 2.6. Analytical Methods of In-House Antibody Characterization

SEC-HPLC (SEC: size exclusion chromatography; HPLC, high-performance liquid chromatography) equipment (Shimadzu LC 20 system, SHIMADZU, Kyoto, Japan) was utilized with a 5 µm, 7.88 mm, and I.D. 300 nm column (TOSOH TSKgel G3000SW, TOSOH Bioscience, Tokyo, Japan), utilizing in-house antibodies and a commercially available monoclonal antibody (11E59, Cat No: 9056, Median Diagnostic, Chuncheon, Republic of Korea) which specifically targets the PCV2 capsid protein. The commercial monoclonal antibody was used to verify the specificity and performance of the purified in-house polyclonal antibodies (rabbit anti-PCV2d IgG) in the in-house ELISA system. The running buffer (A) was used, with a flow rate of 1 mL/min for 20 min, and the sample concentration was set to 1.0 mg/mL. Peak analysis was conducted at 214 nm and 280 nm to assess the size distribution of the antibody.

To characterize the protein profiles, the purified antibodies were analyzed regarding their molecular size using SDS-PAGE (sodium dodecyl sulfate–polyacrylamide gel electrophoresis) to identify the IgG monomer, heavy chain, and light chain. Antibodies extracted and purified from rabbits are typically known to have a molecular weight of approximately 150 kDa, with two heavy chains of about 50 kDa each and light chains of about 25 kDa [43]. Electrophoresis was performed using an 8% polyacrylamide gel.

### 2.7. Construction and Standardization of an In-House PCV2d Indirect ELISA

The in-house PCV2d ELISA was developed as an indirect ELISA due to its distinct advantages. This format reduces costs by requiring only a single labeled secondary antibody, making it efficient for large-scale validation studies. Additionally, the indirect ELISA offers flexibility by allowing the use of unlabeled primary antibodies, facilitating the detection of antibodies from various host species, such as guinea pigs and pigs. Its reliability in quantifying antibody levels further supports its application in serological studies and preclinical vaccine research.

#### 2.7.1. Standard Curve

A standard curve for the in-house ELISA was generated by coating the ELISA plate with antigens at concentrations of 10^4^, 10^5^, and 10^6^ genomic copies/mL. The in-house antibody (WG-PCV2d pAb, rabbit anti-PCV2d IgG) was subjected to 10-fold serial dilutions, ranging from 1 mg/mL to 1 ng/mL. To evaluate blocking efficiency, the plate was blocked with skim milk or BSA at concentrations of 0.1%, 1%, 3%, and 5% in 0.05% PBS-T (PBS-Tween20) (Tween20, Sigma-Aldrich, St. Louis, MO, USA). The primary (WG-PCV2d pAb) and enzyme-labeled secondary antibody (Goat Anti-Rabbit IgG HRP, GTX213110-01, GeneTex, Alton Pkwy Irvine, CA, USA) dilution buffers were prepared according to the blocking agent used, with 1% skim milk in 0.05% PBS-T or 1% BSA in 0.05% PBS-T. Following a 5 min reaction with the TMB substrate, the reaction was stopped using sulfuric acid, and optical density (OD) values were measured at 450 nm to establish the standard curve. Standard curves were generated for each condition to evaluate the relationship between antibody concentration and absorbance under the different blocking and coating conditions.

#### 2.7.2. LOD (Limited of Detection)

The limit of detection (LOD) was determined under the same conditions used to generate the standard curve. LOD represents the lowest concentration of the in-house antibody that can be reliably distinguished from the background signal. The LOD was calculated using the following formula:$$LOD=*Bmean\left(Blank\;mean\;OD\right)+3*\#\sigma B(Stnadarad\;Deviation)$$where Bmean is the mean background signal (OD value from wells without the antibody) and σB is the standard deviation of the background signal. The LOD was calculated for each blocking condition (skim milk and BSA) and antigen concentration, ensuring precision across varying experimental setups.

#### 2.7.3. ROC (Receiver Operating Characterization)

ROC curve analysis was performed to assess the diagnostic performance of the in-house ELISA. True positive and false positive rates were calculated for each antibody concentration using the OD values. The area under the ROC curve (AUC) was determined to evaluate the assay’s overall sensitivity and specificity.

#### 2.7.4. SNR (Signal-to-Noise Ratio)

The signal-to-noise ratio (SNR) was calculated to quantify the assay’s performance in differentiating specific signals from the background noise. The SNR was computed using the following formula:$$SNR=\frac{Mean\;signal\;Intensitiy}{Mean\;Blank\;Intensity}$$

The mean signal OD was derived from wells with the highest antibody concentration, while the mean background OD was obtained from wells without antibodies. SNR values were compared across the different blocking conditions (BSA and skim milk) to identify the most effective blocking agent for minimizing background noise.

#### 2.7.5. Establish the Cut-Off Value of an In-House ELISA

The assay was performed by coating the ELISA plates with antigens at a concentration of 10^5^ genomic copies/mL and blocking with 5% BSA in PBST. The antibody dilution was prepared using 1% BSA in PBST.

The cut-off value for each ELISA was calculated based on the following formula:

$$Cut-off\;Value=*Bmean\left(Blank\;mean\;OD\right)+2*\#\sigma B(Stnadarad\;Deviation)$$ where*Bmean: The mean of the blank’s OD;#σB: The standard deviation of the blank.

#### 2.7.6. Precision

The precision of the in-house ELISA was evaluated under controlled experimental conditions. The ELISA plates were coated with antigens at concentrations of 10^4^, 10^5^, and 10^6^ genomic copies/mL. Blocking was performed using 5% BSA or 5% skim milk. The primary antibody was used at a fixed concentration of 100 ng/mL, and the secondary antibody was diluted at a ratio of 1:2000 in the dilution buffer (1% BSA or 1% skim milk in PBST). Precision was assessed by calculating the intra-assay and inter-assay coefficient of variation (CV) based on replicate OD measurements under identical conditions. The formulas used are as follows:

[Intra-assay CV]

The intra-assay precision was calculated to assess the variability within the same assay plate:

$$Intra\;CV\left(\%\right)=\frac{*\sigma intra}{\#µintra}*100$$ where*σintra: The standard deviation of replicate OD values within a single plate;#μintra: The mean OD value of the replicates within the plate.

[Inter-assay CV]

The inter-assay precision was calculated to evaluate variability between multiple plates:

$$Inter\;CV\left(\%\right)=\frac{*\sigma inter}{\#µinter}*100$$ where *σintra: The standard deviation of mean OD values from multiple plates;#μintra: The overall mean OD value across plates.

#### 2.7.7. Analytical Specificity via a Competitive Binding Assay

The analytical specificity of the in-house ELISA was evaluated through a competitive binding assay using heterologous viral antigens (PCV2a, PCV2b, and PRRSV). ELISA plates were coated with the target antigen (inactivated PCV2d whole virus, 10^5^ genomic copies/mL) under optimized conditions (5% BSA blocking agent, 1% BSA in PBST for antibody dilution, and 100 ng/mL of the in-house antibody (primary antibody)).

Competitive inhibition was simulated by co-incubating the primary antibody with either homologous (PCV2d) or heterologous antigens (PCV2a, PCV2b, and PRRSV) at two competition ratios (1:1 and 1:5, antigen/antibody, *w*/*w*). Signal output (OD at 450 nm) was measured following the TMB reaction and used to assess specificity. Specificity was inferred based on the absence of significant OD signal suppression in the presence of non-target antigens, indicating minimal cross-reactivity.

#### 2.7.8. Robustness

The antigen was coated at a concentration of 10^5^ genomic copies/mL, and blocking was performed using either 5% BSA or 5% skim milk at concentrations of 0.5%, 1%, 3%, and 5%. The primary and secondary antibody concentrations were consistent with the previously described conditions (1 µg/mL for the primary antibody and 1:2000 for the secondary antibody). The TMB substrate was utilized for three different time intervals: 3 min, 5 min, and 10 min.

#### 2.7.9. Validation Through Comparison with a Commercialized ELISA Kit

To validate the performance of the in-house ELISA for detecting anti-PCV2d IgG, comparative analyses were performed with two commercial kits: Ingezim Circo IgG^®^ (Company I, Gold Standard Diagnostic, Budapest, Hungary) and Bionote Circo Ab ELISA 2.0^®^ (Company B, Bionote, Hwaseong, Republic of Korea). For anti-PCV2d IgG detection using the in-house ELISA, the goat anti-guinea pig IgG (H + L) secondary antibody conjugated to horseradish peroxidase (HRP) (#A18769, Invitrogen, Waltham, MA, USA) was used for guinea pig sera, and the goat anti-pig IgG H&L (HRP) antibody (#ab102135, Abcam, London, UK) was used for pig sera.

#### 2.7.10. Data Analysis

Statistical analyses and graphing were performed using GraphPad Prism 9.0 (GraphPad Software, Boston, MA, USA) for ROC curves, regression modeling, and significance testing, and SPSS Statistics 17.0 (IBM, Armonk, NY, USA) was used for inter-group comparisons and CV analysis. Four-parameter logistic regression was used for standard curve fitting, and one-way ANOVA with Tukey’s post hoc test or paired *t*-tests were applied as appropriate. Distinct letters or asterisks indicate statistically significant differences (*p* < 0.05).

## 3. Results

### 3.1. Phylogenetic Analysis of IMMUNIS^®^ DMVac

Based on the capsid protein gene of PCV2 (ORF 2 (open reading frame 2)), three well-defined groups that correspond to each PCV2 were established (Figure 2). A genetic distance of approximately 0.05–0.08% has been observed among the subtypes of PCV2 (PCV2a, PCV2b, and PCV2d). Additionally, PCV2 strains in South Korea were identified to be genetically close to those reported in neighboring countries, including China. The PCV2d strain (MW623084 SNUVR201901) used in the DMVac vaccine was found to form a distinct subtree, differing from previously isolated PCV2d strains.

### 3.2. Characterization of the In-House Antibody (Rabbit Anti PCV2d IgG)

The antiserum which was produced by a rabbit experiment was purified using a HiTrap™ Protein A HP column (Cytiva, USA) with the ÄKTA pure™ system. Elution was performed using buffer B following equilibration with buffer C. The polyclonal IgG fraction was eluted, corresponding to the main antibody peak (Appendix A). For quantification, a standard curve was generated based on known IgG concentrations, exhibiting a high degree of linearity (R^2^ = 0.9992) (Appendix A). To assess the structural integrity and purity of the prepared antibody, size exclusion chromatography–high performance liquid chromatography (SEC-HPLC) was conducted. The in-house IgG appeared as a sharp monomeric peak with a retention time of 7.996 min and a peak area of 95,091, indicating high monomeric purity without evidence of aggregation or degradation (Appendix A). Additionally, SDS-PAGE was performed to evaluate possible interference from the elution and equilibration buffers. No significant background signals or protein aggregation artifacts were observed under the denaturing conditions, confirming the antibody’s structural stability and suitability for use in downstream immunoassays.

### 3.3. Optimization of Key ELISA Parameters: The Blocking Agent, Antigen Coating, and Substrate Reaction Time

To refine the performance of the in-house ELISA, we systematically evaluated three major parameters: the blocking agent composition (BSA at 0.5–5%), the coating antigen concentration (10^4^–10^6^ genomic copies/mL), and the substrate reaction time (3, 5, and 10 min). The highest diagnostic accuracy (AUC = 0.97) was observed for the combination of 10^5^ FAID_50_/mL of the coating and a 5 min substrate reaction, which was consistent across all BSA concentrations (Figure 3A).

Subsequent analysis comparing BSA concentrations under these optimized conditions showed that BSA 5% produced the most stable diagnostic performance, achieving the highest AUC value (0.99) with minimal signal distortion and a consistent dynamic range (Figure 3B). Additional ROC curves across other BSA concentrations (0.5%, 1%, and 3%) are presented in Appendix A for completeness.

While skim milk-based blocking was also tested under identical conditions (Appendix A), BSA provided higher overall reproducibility, sharper titration curves, and lower background noise, particularly at lower antigen concentrations.

### 3.4. Precision Evaluation of the In-House ELISA

Intra- and inter-assay precision of the in-house ELISA was assessed using two blocking conditions (5% BSA and 5% skim milk), with antigen coating concentrations of 10^4^ and 10^5^ genomic copies/mL. As summarized in Table 1, all BSA-blocked conditions satisfied the acceptance criteria of <10% for intra-assay CV and <15% for inter-assay CV.

In contrast, multiple conditions using 5% skim milk as the blocking agent, particularly at the 1:8 and 1:16 antibody dilutions, exhibited intra-assay CVs approaching the upper limit of the acceptable threshold, with values reaching up to 13.8%. Despite this variability in intra-assay repeatability, inter-assay CVs remained within the predefined acceptance criterion (<15%) across all tested conditions. These findings suggest that 5% BSA provided greater intra- and inter-assay consistency than skim milk, supporting its suitability as a more reliable blocking agent under the current assay configuration.

### 3.5. Specificity and Signal Responsiveness of the In-House ELISA Under Optimized Conditions

To validate the constructed conditions of the in-house ELISA (10^5^ genomic copies/mL of the coating antigen, 5% BSA or the skim milk blocking agent, 100 ng/mL of the in-house antibody (1st antibody), and 5 min substrate incubation), a series of experiments was conducted to evaluate signal responsiveness, blocking efficiency, and analytical specificity.

OD values demonstrated a strong inverse linear correlation with antigen dilution, confirming the assay’s quantitative responsiveness across the working range (Figure 4A, R^2^ = 0.9890).

When comparing blocking agents, both 5% BSA and skim milk enabled distinct signal resolution across serial antigen dilutions (Figure 4B); however, BSA provided superior signal-to-background ratios and minimized the nonspecific background, particularly at mid-range coating concentrations (1:16–1:64).

Analytical specificity was further evaluated by competitive ELISA using heterologous antigens (PCV2a, PCV2b, and PRRSV). The in-house antibody showed no appreciable reduction in OD values in the presence of these non-target antigens at ratios of 1:1 and 1:5 (Figure 4C–E), indicating a lack of cross-reactivity and confirming high analytical specificity under the tested conditions. However, as the in-house ELISA utilizes a whole-virus immunogen and the capsid protein among PCV2 genotypes shares over 90% sequence identity, further competition assays using homologous PCV2d antigens are warranted to confirm type-specific binding. These complementary analyses are planned as part of future validation studies to address potential epitope-level cross-reactivity.

### 3.6. Comparative Diagnostic Performance of In-House and Commercial ELISA Kits

At 21 days post-vaccination (DPV), the in-house ELISA demonstrated a vaccinated period and dose-dependent increase in anti-PCV2d IgG S/P ratios in vaccinated guinea pigs (Figure 5A), showing statistically significant distinctions between the quarter, half, and full dose groups (*p* < 0.05). The temporal and dose-responsive trends observed with the in-house ELISA were consistent with those obtained using the commercial kits (Figure 5C,E), indicating diagnostic concordance across platforms and supporting its applicability for antibody monitoring.

Field serum samples collected from 2022 to 2024 were categorized by consensus of two commercial assays (positive, false positive, and negative) and subsequently analyzed using the in-house ELISA (Figure 5B). The assay showed high discriminatory accuracy across all three categories, with false positive rates lower than those observed in Company B (Figure 5F). While the commercial kits were applied according to the manufacturers’ instructions, including predefined cut-off values (Company I: 0.376; Company B: 0.4 [Figure 5D]), the in-house ELISA employed a cut-off value of 0.4, which was established through prior experimental validation. All three assays detected comparable classification patterns across positive, false positive, and negative field samples, with consistent trend alignment observed over the 2022–2024 period.

### 3.7. Consistency and Predictive Validity of the In-House ELISA Compared to Commercial Kits and Porcine Sera

To evaluate the analytical consistency and interspecies predictive validity of the in-house ELISA, we compared its performance against two commercial diagnostic kits (Company I and Company B) and porcine field sera (2022–2024) under optimized assay conditions (10^5^ genomic copies/mL of the coating antigen, the 5% BSA blocking agent, 100 ng/mL of the in-house antibody (rabbit anti-PCV2d IgG, primary antibody), the appropriate enzyme-labeled antibody (secondary antibody against guinea pig and porcine antibodies, respectively), and 5 min substrate incubation). As shown in the estimation plot (Figure 6A), the S/P ratios obtained from the in-house ELISA were statistically comparable to those from both commercial kits. No significant differences were observed between paired measurements, and the mean differences remained within acceptable variation limits, supporting the assay’s intra- and inter-assay agreement.

To further assess the cross-species predictive potential, antibody responses in guinea pig antisera, which were generated using the same vaccine batch, were compared with those from porcine field sera. A total of 14 matched data points were generated by pairing each guinea pig S/P value with the corresponding pig serum tested under identical in-house ELISA conditions. Linear regression analysis revealed a modest but positive correlation (R^2^ = 0.1815; Figure 6B), suggesting a moderate but interpretable relationship between the two species’ humoral responses.

Complementary Bland–Altman analysis demonstrated a mean bias of −0.21 and 95% limits of agreement ranging from −0.48 to 0.07 (Figure 6C), indicating acceptable agreement between guinea pig and porcine S/P values. Although variation was observed at higher titer ranges, most values remained within the limits of agreement, supporting the technical feasibility of using guinea pig antisera as a reference indicator for porcine antibody levels. These results suggest that the in-house ELISA detects consistent and comparable outputs across species under standardized conditions.

## 4. Discussion

This study aimed to establish a scientifically standardized in-house ELISA for detecting anti-PCV2d IgG antibodies, addressing the limitations of the current T/C-based criteria mandated by the Animal and Plant Quarantine Agency (APQA) in the Republic of Korea. While the T/C method—based on dividing the OD of the test group by the mean OD of the control group—has been widely used, it lacks a statistical foundation and is not optimal for assessing true seroconversion or vaccine-induced antibody responses [44,45]. Our approach focused on the development of an indirect ELISA platform using in-house polyclonal antibodies generated from rabbits immunized with the whole inactivated PCV2d virus, followed by a systematic optimization and analytical validation process [33,34,39].

The in-house ELISA displayed high diagnostic performance under defined assay conditions, including 10^5^ genomic copies/mL of the antigen coating, the 5% BSA blocking agent, 100 ng/mL of the in-house antibody (rabbit anti PCV2d, primary antibody), and 5 min TMB incubation. ROC curve analysis based on antisera generated from guinea pigs immunized with the virus-cultured PCV2d antigen demonstrated an AUC of 0.97, indicating high diagnostic sensitivity and specificity in differentiating vaccinated from various animals. Linearity tests confirmed that OD values decreased proportionally with antigen dilution, with R^2^ = 0.9890 under optimal conditions, supporting the assay’s quantitative responsiveness [46]. Blocking agents were systematically compared, and BSA consistently resulted in lower background OD values, improved signal-to-noise ratios, and superior intra-/inter-assay reproducibility compared to skim milk, consistent with previous PCV2d-specific immunoassay optimization studies [47].

Specifically, all tested conditions using BSA achieved CVs below the accepted thresholds (<10% intra-assay and <15% inter-assay) [48], while some skim milk-based conditions showed increased variability, especially at mid-level antibody dilutions.

Analytical specificity was assessed through a competitive ELISA using heterologous antigens—PCV2a, PCV2b, and PRRSV—and the in-house antibody showed no measurable signal reduction at either 1:1 or 1:5 competition ratios, confirming the platform’s robustness and lack of cross-reactivity [49]. While these results support analytical specificity under the current assay conditions, it is acknowledged that the high sequence identity among PCV2 genotypes (over 90%) may limit definitive type-specific antibody discrimination. Therefore, homologous competition assays using PCV2d antigens will be incorporated in future studies to more directly validate epitope specificity and ensure genotype-resolved antibody–antigen interaction. In addition, the specificity of the in-house ELISA was assessed against PCV2a, PCV2b, and PRRSV, and further studies are warranted to evaluate potential cross-reactivity with other unrelated pathogens such as PEDV (Porcine epidemic diarrhea virus), SIV (swine influenza virus), *Mycoplasma hyopneumoniae*, and *Actinobacillus pleuropneumoniae*, which are common co-infecting pathogens in swine herds.

Comparative evaluation with two commercial ELISA kits demonstrated strong diagnostic concordance. Time-course analysis in guinea pigs vaccinated with a quarter, half, or full dose of the PCV2d vaccine revealed consistent S/P ratio patterns across the in-house and commercial assays. Notably, all assays detected seroconversion by 14–21 DPV, with no significant differences in group-wise S/P values. Moreover, field samples collected from 2022 to 2024 were classified into positive, false positive, and negative groups by commercial kit consensus and tested with the in-house ELISA. All three kits showed comparable classification trends over the years, affirming the reproducibility and field applicability of the in-house platform. Importantly, the in-house ELISA employed a cut-off value of 0.4, which was established through prior statistical analysis based on the blank and negative control distributions, rather than adopting manufacturer-defined or conventionally assumed thresholds [50]. This cut-off value aligned well with those used in commercial kits (Company I: 0.376; Company B: 0.4) and demonstrated clear group separation, particularly in recent isolates.

A comparative analysis of S/P ratios derived from guinea pig and pig sera using the in-house ELISA was conducted to evaluate cross-species predictive validity. Linear regression analysis revealed a modest but positive correlation (R^2^ = 0.1815), reflecting a consistent directional trend despite expected interspecies variation in the immunoglobulin repertoire and antibody maturation.

To further examine the agreement between guinea pig and porcine antibody responses, Bland–Altman analysis was performed on 14 matched S/P value pairs. The mean bias was −0.21, and the 95% limits of agreement ranged from −0.48 to 0.07. Most values fell within this range, suggesting a quantitatively acceptable level of agreement [51]. Although variability was observed at higher antibody titers, the overall agreement and positive correlation between guinea pig and porcine sera support the feasibility of using guinea pig antisera as a practical surrogate for serological monitoring in pigs under standardized ELISA conditions. This approach is consistent with previous studies validating interspecies ELISA-based comparisons for antibody detection [52].

Collectively, these findings indicate that the in-house ELISA platform provides reliable and reproducible results across species, offering utility not as a diagnostic substitute but as a scientifically justified tool for evaluating immunogenicity during vaccine production or regulatory verification [53]. This level of correlation is not unexpected, considering intrinsic interspecies variation in immune response dynamics, including differences in antibody repertoire maturation, isotype switching, and epitope-specific binding profiles [54].

Collectively, the findings of this study support the use of the in-house ELISA as a reproducible, scalable, and diagnostically reliable platform for detecting PCV2d-specific IgG antibodies. The assay was considered to have appropriate core validation criteria, including analytical sensitivity, specificity, intra- and inter-assay precision, and robustness, under experimentally defined and optimized conditions. Compared to the currently adopted T/C-ratio-based evaluation system, which lacks a statistical basis for threshold setting and standardization, the in-house ELISA provides a scientifically validated and transparent alternative for serological evaluation in guinea pig models [55].

Taken together, these findings provide scientific evidence that the in-house ELISA fulfills essential performance requirements for the serological evaluation of PCV2d vaccines. By demonstrating analytical validity under controlled conditions—including antigen-specific precision, diagnostic specificity, and platform robustness—the assay offers a credible alternative to the current T/C-ratio-based system. Importantly, the methodological transparency and reproducibility of this assay support its alignment with regulatory expectations and reinforce its potential utility as a scientifically justified tool within the Korean APQA vaccine evaluation framework. Accordingly, the adoption of this ELISA format may contribute to improved standardization, reproducibility, and scientific rigor in the domestic veterinary vaccine licensing and development process.

## 5. Conclusions

This study established and validated a statistically defined in-house ELISA for the detection of PCV2d-specific IgG, not as a commercial diagnostic tool but as an internally standardized platform for verifying immunogenicity during vaccine production. The assay demonstrated high diagnostic performance, with excellent sensitivity and specificity (AUC = 0.97), strong linearity (R^2^ = 0.9890), and reproducible intra-/inter-assay precision under optimized assay conditions. Comparative evaluations with two commercial kits confirmed diagnostic equivalence in detecting seroconversion and classifying field samples from 2022 to 2024. Additionally, cross-species analyses showed a positive correlation between guinea pig and porcine sera (R^2^ = 0.1815), and Bland–Altman analysis revealed acceptable agreement (mean bias = −0.21 and 95% limits of agreement = −0.48 to 0.07). These results suggest that guinea pig antisera, which were generated using the same vaccine bulk, may serve as a scientifically valid reference indicator for evaluating vaccine immunogenicity in swine. The in-house ELISA incorporated a statistically derived cut-off value and demonstrated antigen specificity, platform robustness, and field applicability.

Collectively, these findings support the assay’s utility as a standardized immunogenicity evaluation tool aligned with the Animal and Plant Quarantine Agency (APQA)’s lot-release requirements. Its implementation may contribute to enhanced transparency, reproducibility, and scientific rigor in Korea’s veterinary vaccine quality control and regulatory approval processes.

## Figures and Tables

**Figure 1 vaccines-13-00657-f001:**
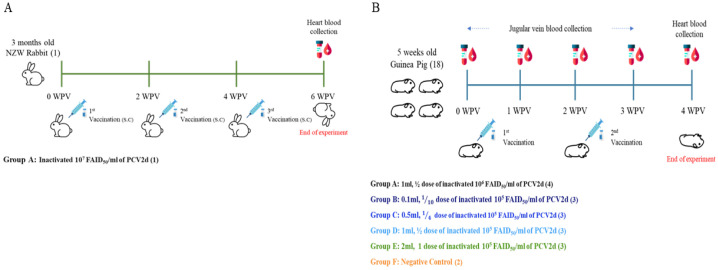
**Overview of the animal experiment.** (**A**) One 3-month-old specific-pathogen-free (SPF) New Zealand White (NZW) rabbit was used for in-house antibody production. The rabbit was inoculated three times with vaccinations at two-week intervals subcutaneously (S.C.). Daily monitoring was performed to detect any adverse reactions following vaccination, and the final antiserum was obtained after euthanasia. (**B**) Eighteen 5-week-old SPF guinea pigs were randomly divided into five groups. The vaccine group was immunized following the experimental design, receiving intramuscular (I.M.) injections at two-week intervals (0 WPV and 2 WPV). Animals were monitored daily to assess any adverse effects related to vaccination. To confirm antiserum production, a total of five blood samples were collected. The first four samples (0 WPV to 3 WPV) were drawn from the jugular vein, and the final sample was obtained via cardiac puncture after euthanasia.

**Figure 2 vaccines-13-00657-f002:**
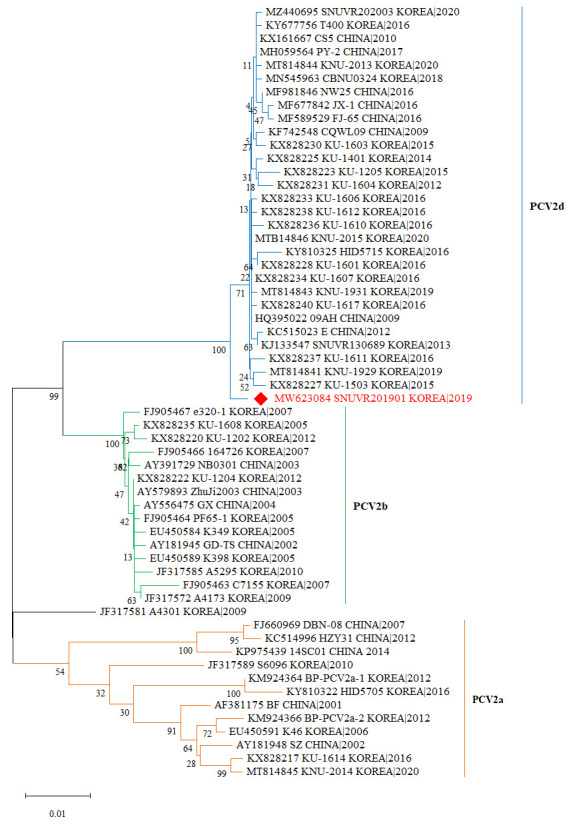
**Phylogenetic analysis.** Phylogenetic trees of PCV2 ORF2 sequences were constructed using the maximum likelihood method based on the generalized time-reversible (GTR) model with G + I in Mega 11. Bootstrap values were calculated with 1000 replicates. The red diamond indicates the PCV2d strains of IMMUNIS^®^ DMVac in this study.

**Figure 3 vaccines-13-00657-f003:**
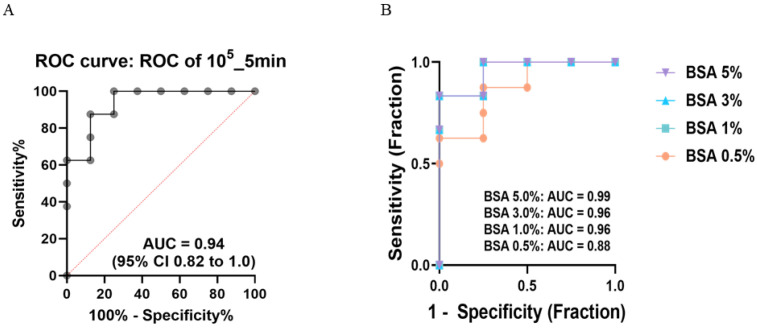
**Dual-parameter optimization of ELISA diagnostic performance and assay linearity in BSA conditions.** Optimization of ELISA diagnostic performance using ROC curve analysis. (**A**) ROC curve under the optimized conditions of the 10^5^ FAID_50_/mL coating concentration and a 5 min substrate reaction, resulting in an AUC of 0.94 (95% CI: 0.82–1.00). (**B**) Comparative overlay of ROC curves for different BSA concentrations (0.5%, 1%, 3%, and 5%) under the same coating and incubation conditions. BSA 5% yielded the highest AUC (0.99), indicating superior diagnostic stability.

**Figure 4 vaccines-13-00657-f004:**
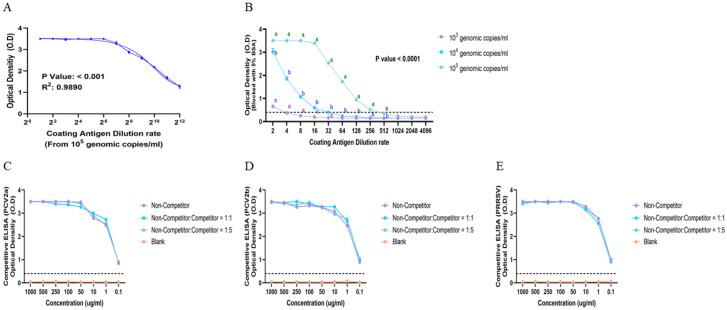
**Evaluation of the in-house ELISA under optimized assay conditions.** (**A**) Linear decrease in OD values across serial dilutions of the coating antigen (10^5^ genomic copies/mL). (**B**) Signal profiles under blocking with 5% BSA; OD differences were statistically significant between antigen dilutions (*p* < 0.0001). (**C**–**E**) Competitive ELISA using heterologous antigens to assess analytical specificity: PCV2a (**C**), PCV2b (**D**), and PRRSV (**E**). No measurable cross-reactivity or signal inhibition was observed at the 1:1 and 1:5 competition ratios.Different letters (a, b, c) indicate statistically significant differences between groups (*p* < 0.05).

**Figure 5 vaccines-13-00657-f005:**
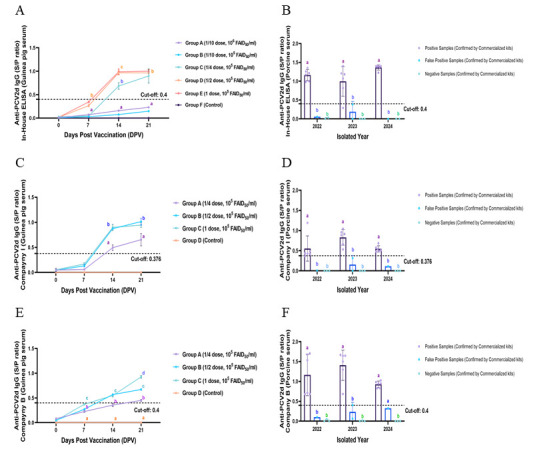
**Comparative analysis of the in-house ELISA and two commercial kits for anti-PCV2d IgG detection.** (**A**,**C**,**E**) S/P ratio profiles in guinea pig sera at 0, 7, 14, and 21 DPV following a 1/4, 1/2, or 1 dose of the PCV2d vaccine, measured by the in-house ELISA (**A**), Company I (**C**), and Company B (**E**). (**B**,**D**,**F**) Field sera (2022–2024) were categorized as positive, false positive, or negative by commercial kit consensus and tested using the in-house ELISA (**B**), Company I (**D**), and Company B (**F**). Commercial cut-off values were applied per manufacturer instructions (Company I: 0.376; Company B: 0.4), while the in-house ELISA used a validated cut-off value of 0.4. All assays showed consistent classification trends. Statistical analysis was performed using one-way ANOVA with Tukey’s post hoc test (*p* < 0.05). Different letters (a, b, c) indicate statistically significant differences between groups (*p* < 0.05).

**Figure 6 vaccines-13-00657-f006:**
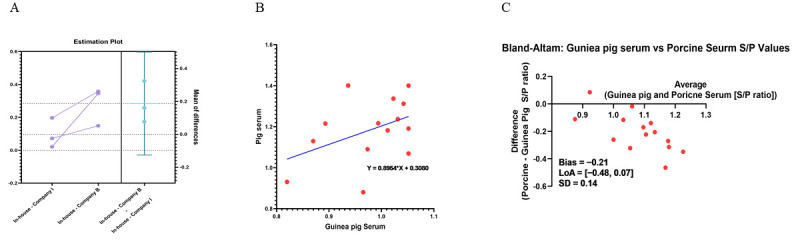
**Comparative analysis of the in-house ELISA with commercial kits and porcine serum samples.** (**A**) Estimation plot comparing S/P ratios from the in-house ELISA and two commercial kits (Company I and B) using guinea pig sera. No significant differences were detected. (**B**) Simple linear regression of S/P values from guinea pig and pig sera tested by the in-house ELISA, showing a positive correlation (R^2^ = 0.1815). (**C**). Bland–Altman plot comparing guinea pig and porcine S/P values, showing a mean bias of −0.21 and 95% limits of agreement ranging from −0.48 to 0.07. All assays were performed under validated conditions (10^5^ genomic copies/mL of the antigen coating, the 5% BSA blocking agent, 100 ng/mL of the in-house antibody (rabbit anti-PCV2d IgG, primary antibody), the appropriate enzyme-labeled antibody (secondary antibody against guinea pig and porcine antibodies, respectively), and 5 min substrate incubation).

**Table 1 vaccines-13-00657-t001:** Precision assay of the in-house ELISA.

Blocked with 5% BSA									
Antigen Coating (Copies/mL)	1st Antibody Dilution (From 100 ng/mL)	Plate 1 Mean O.D.	Plate 2 Mean O.D.	Plate 1 SD	Plate 2 SD	Plate 1 Intra-Assay CV (%)	Plate 2 Intra-Assay CV (%)	Inter-Assay CV (%)	Precision Met (Intra/Inter)
**10^4^**	1:2	3.2175	2.972	0.0771	0.2744	2.395	2.094	2.245	O/O
	1:4	1.812	2.0145	0.1768	0.2305	9.756	2.913	6.335	O/O
	1:8	1.011	1.148	0.0191	0.0368	1.914	3.203	2.558	O/O
	1:16	0.623	0.603	0.0354	0.0057	5.675	0.938	3.307	O/O
**10^5^**	1:16	3.3625	3.433	0.0700	0.0094	2.082	2.760	2.421	O/O
	1:32	2.4395	2.639	0.0007	0.0381	0.029	1.447	0.738	O/O
	1:64	1.7225	1.7485	0.0544	0.1110	3.161	6.349	4.755	O/O
	1:128	0.9075	0.9895	0.0657	0.0869	7.246	8.790	8.018	O/O
	1:256	0.518	0.5275	0.0026	0.0077	5.187	1.475	3.331	O/O
**Blocked with 5% Skim milk**									
**Antigen Coating** **(Copies/mL)**	**1st Antibody dilution** **(From 100 ng/mL)**	**Plate 1** **Mean O.D.**	**Plate 2** **Mean O.D.**	**Plate 1** **SD**	**Plate 2** **SD**	**Plate 1** **Intra-assay** **CV (%)**	**Plate 2** **Intra-assay** **CV (%)**	**Inter-assay** **CV (%)**	**Precision Met** **(Intra/Inter)**
**10^4^**	1:2	2.675	2.9485	0.1032	0.3345	3.3859	11.343	7.601	X/O
	1:4	1.6585	1.613	0.0530	0.1103	3.198	6.839	5.018	O/O
	1:8	0.7885	0.854	0.0049	0.0410	0.628	4.802	2.715	O/O
	1:16	0.4545	0.4045	0.0629	0.0474	13.847	11.712	12.779	X/O
**10^5^**	1:16	3.2915	3.3715	0.0728	0.0530	2.213	1.573	1.893	O/O
	1:32	2.266	2.109	0.2065	0.3154	9.112	14.954	12.033	X/O
	1:64	1.2525	1.267	0.1209	0.0014	9.654	0.112	4.833	O/O
	1:128	0.5385	0.634	0.0728	0.0141	13.525	2.231	7.878	O/O

SD: Standard deviation. CV: Coefficient of variation. “O” = met precision criteria; “X” = did not meet precision criteria. The first symbol refers to intra-assay precision, and the second to inter-assay precision.

## Data Availability

The data that support the findings of this study are available from the corresponding author upon reasonable request.

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
