# Peer review of "An In-House ELISA for Anti-Porcine Circovirus Type 2d (PCV2d) IgG: Analytical Validation and Serological Correlation"

_vaccines, 2025, doi:10.3390/vaccines13060657_

Round 1

Reviewer 1 Report

Comments and Suggestions for Authors

In this study, the authors developed a statistically defined in-house ELISA for the detection of PCV2d-specific rabbit IgG, possibly offering a scientifically substantiated alternative to currently adopted system for pigs in Korea. Although they indicated the usefulness of the ELISA using field pig sera, they should reevaluate experimental conditions, especially blocking solution, using pig serum. There are also many errors or insufficient explanations in the manuscript. Below are the  examples: 

  1. Title. Anti PCV2d antibody detection not PCV2d detection.
  2. Line 57. Four not eight?
  3. Line 92. What is FAID? What is the relation to FFU or genomic copy?
  4. Line 121. 18 not 20?
  5. Line 126. 4-week-old?
  6. Line 142. Three times?
  7. Line 156. What was administrated into the control? PBS or PBS+adjuvant?
  8. Line 183. TOSOH not TOSHO.
  9. Line 338. SDS-Page is not native gel electrophoresis.
  10. Lines 493-495. The authors did not check unrelated swine pathogens.
  11. The authors also indicated no secondary antibodies against guinea pig and pig sera.
Comments on the Quality of English Language

Correction of the paper by a native English speaker is recommended.

Author Response

  1. Anti PCV2d antibody detection not PCV2d detection.
    • Thank you for your valuable suggestion. We agree that the original title may have been misleading, as the aim of the study is to validate an ELISA for detecting antibodies against PCV2d, rather than detecting the virus itself. Accordingly, we have revised the title to more accurately reflect the scope and purpose of the study. The updated title is:

Validation and Correlation Analysis of an In-House ELISA for Anti-PCV2d IgG Detection Using Biological Samples and Virus Culture Samples. We appreciate your insightful comment, which has helped improve the clarity and precision of the manuscript.

  1. Line 57. Four not eight?
    • Thank you for pointing this out. The original sentence listed only five genotypes while referring to eight, which was inconsistent and potentially confusing. We have revised the sentence to correctly reflect all eight currently recognized PCV2 genotypes. The corrected sentence now reads (Line: 60-61)

"From a taxonomic perspective, PCV2 has been classified into eight genotypes: PCV2a, PCV2b, PCV2c, PCV2d, PCV2e, PCV2f, PCV2g, and PCV2h." We appreciate your attention to detail.

  1. Line 92. What is FAID? What is the relation to FFU or genomic copy?
    • Thank you for raising this important point. FAID50 stands for fluorescence active infectious dose 50%, a titer unit based on quantifying fluorescently detectable viral infection in host cells.

This approach has been previously reported in viral quantification studies, such as for SFTSV (Park, Seok-Chan, et al. "Pathogenicity of severe fever with thrombocytopenia syndrome virus in mice regulated in type I interferon signaling: Severe fever with thrombocytopenia and type I interferon." Laboratory Animal Research 36 (2020): 1-10.), and was adopted in our study as a reliable method for determining PCV2d infectivity. Although FAID50 is not directly interchangeable with FFU or genome copies, it reflects a biologically active virus titer and is periodically calibrated against standard TCID50 assays. We have clarified this in the revised manuscript (Line 103-108).

  1. Line 121. 18 not 20?
    • Thank you for pointing this out. You are correct—the correct number of guinea pigs used in the experiment was 18, not 20. We apologize for the oversight and have corrected the sentence in the revised manuscript to: (Line: 140)

“The animal experiments were performed on 1 rabbit and 18 guinea pigs.”

We appreciate your careful review.

  1. Line 126. 4-week-old?
    • Thank you for your attention to detail. The correct age of the rabbit used in the experiment was 3 months, not 4 weeks. We apologize for the error and have revised the sentence accordingly in the manuscript: (Line: 145)

“One 3-month-old female New Zealand White (NZW) rabbit was purchased...”

We appreciate your careful review and have made the necessary correction.

  1. Line 142. Three times?
    • Thank you for your comment. You are correct—the original sentence lacked clarity. The animals were indeed immunized three times at 2-week intervals, as intended in the experimental design. We have revised the sentence for clarity as follows: (Line 165)

“The inoculation was administered three times at 2-week intervals.”

We appreciate your careful review and have confirmed that this information is now correctly stated in the manuscript.

  1. Line 156. What was administrated into the control? PBS or PBS+adjuvant?
    • Thank you for your comment. The control group (Group F) was administered PBS only, without any adjuvant, to serve as a negative control. We have revised the sentence in the manuscript to clarify this point: (Line 181)

“Group F (control) was administered PBS only, without adjuvant.”

We appreciate your thoughtful observation.

  1. Line 183. TOSOH not TOSHO.
    • Thank you for pointing out the typographical error. We have corrected the spelling from “TOSHO” to the correct manufacturer name “TOSOH” in the revised manuscript (Line: 220).
  2. Line 338. SDS-Page is not native gel electrophoresis.
    • Thank you for your comment. You are absolutely right — the method used was SDS-PAGE, which is a denaturing electrophoresis technique, not native gel electrophoresis. We have corrected the terminology in the manuscript and revised the sentence to: (Line: 371-374)

“Additionally, SDS-PAGE was performed to evaluate possible interference from the elution and equilibration buffers. No significant background signals or protein aggregation artifacts were observed under the denaturing conditions.”

We appreciate your attention to methodological accuracy.

  1. Lines 493-495. The authors did not check unrelated swine pathogens.
    • Thank you for your insightful comment. We agree that evaluating cross-reactivity with a broader range of unrelated swine pathogens would further support the specificity of the in-house ELISA. While our current study assessed specificity against PCV2a, PCV2b, and PRRSV, we acknowledge the importance of including other common co-infecting but unrelated pathogens in future evaluations. To address this point, we have revised the manuscript to include the following statement: (Line 541-548)

“Although the specificity of the in-house ELISA was assessed against PCV2a, PCV2b, and PRRSV, further studies are warranted to evaluate potential cross-reactivity with other unrelated pathogens such as Porcine Epidemic Diarrhea Virus (PEDV), Swine Influenza Virus (SIV), Mycoplasma hyopneumoniae, and Actinobacillus pleuropneumoniae, which are commonly co-infecting agents in swine herds.”

We appreciate your valuable suggestion, which has helped improve the clarity and scope of the discussion.

  1. The authors also indicated no secondary antibodies against guinea pig and pig sera.
    • Thank you for pointing this out. We apologize for the lack of clarity in the original description. In fact, appropriate secondary antibodies were used in the in-house ELISA to detect IgG in both guinea pig and pig sera. Specifically, we used a goat anti-guinea pig IgG (H+L) secondary antibody conjugated to horseradish peroxidase (HRP) (#A18769, Invitrogen, USA) for guinea pig sera, and a goat anti-pig IgG H&L (HRP) antibody (#ab102135, Abcam, UK) for pig sera.

We have revised the manuscript to include these details in the relevant section describing the comparative ELISA analysis (Lines 334–337).

We appreciate your careful review and have corrected the oversight accordingly.

Reviewer 2 Report

Comments and Suggestions for Authors

he manuscript (vaccines-3670852) entitled “Validation and Correlation Analysis of an In-House ELISA for PCV2d Detection Using Biological Samples and Virus Culture Samples” by Gyeong-Seo Park et al. describes the establishment of specific ELISA detecting for PCV2 antiserum. Established ELISA is expected to be used for the quality control of the PCV2 vaccine. The study was conducted in a step-by-step manner and analyzed thoroughly. This manuscript meets the study fields covered by the journal, Vaccines.

This reviewer thinks this study is worth noting in the journal. However, this reviewer has one Major concern about the specificity of this ELISA and several Minor comments that need to be clarified by the authors before accepting for publication.

Major concern

The authors claim that the ELISA system specifically detects antisera against PDV2d because the antibody-antigen reactivity of ELISA is not influenced by PCV2a, PCV2b, or PRRSV antigens. This reviewer thinks this experiment needs more clarification. First of all, there is a high degree of identity and similarity among the capsid proteins of PCV2a, PCV2b, and PCV2d. Identity is 93% between PCV2d SNUVR201901 (MW623084) and PCV2a S9096 (JF317589), and 91% between PCV2d SNUVR201901 (MW623084) and PCV2b e320-1 (FJ905467). Similarity is 99% between PCV2d SNUVR201901 (MW623084) and PCV2a S9096 (JF317589), and 98% between PCV2d SNUVR201901 (MW623084) and PCV2b e320-1 (FJ905467). In these extremely high identity and similarity conditions, it is generally difficult to detect the type-specific antibody using the conventional hole capsid protein. The authors led the conclusion since no competition was found in the presence of PCV2a, PCV2b, and PRRSV antigens (Figure 5D-5F). However, this study lacks a competition experiment to assess the antigen-antibody reaction, as the homologous PCV2d antigen hampers ELISA.

Minor comments

  1. Lines 2-4, Title: Please specify the ELISA target as an anti-PCV2d antibody.
  2. Lines 24-25, Abstract: As shown in the above Major concern, this reviewer thinks this experiment needs positive control to confirm the validity of the competition experiment.
  3. Lines 38-39: Please add the genus name for PCV, like the genus Circovirus family Circoviridae, together with the genus
  4. Line 59: Over time, the predominant genotypes of PCV2 have slowly shifted from PCV2a to PCV2b to PCV2d, as shown in the text. However, views that PCV2d does not escape the immunity induced by PCV2a-based vaccines had better be added to the text [doi:10.1016/j.vetmic.2023.109796, doi.org/10.3390/vetsci6030061].
  5. Line 92: A species name “ hyopneumoniae” should be described as “Mycoplasma hyopneumoniae”.
  6. Line 109: Please explain why the authors used real-time reverse transcription-polymerase chain reaction (RT-qPCR) for detecting DNA viruses such as PCV2. PCV2 needs no reverse transcription before DNA amplification.
  7. Lines 113 and 119: A term ORF2 had better be shown as ORF2
  8. Line 116: Please check if the author really used the DNA assembly software, Lasergene Seqman Pro (DNA Star, USA) for the small DNA virus like PCV2.
  9. Line 131, Section 2.4: Twenty guinea pigs are described to be used here, but eighteen guinea pigs in 6 groups (4+3+3+3+3+2=18) were used in Figure 1B. Please check the number of guinea pigs.
  10. Line 137, Section 2.4.1: One 4-week-old female NZW rabbit is described to use here, but a 3-month-old NZW rabbit is shown to use in Figure 1A. Please check the age of the guinea pigs used for the study.
  11. Line3 138-142: It is described that “The animal experiment protocol was approved by the WOOGENE B&G Co., Ltd Institutional Animal Care and Use Committee (WG-IACUC-2024-004) and performed following the guidelines and regulations detailed by the committee.” (lines 122-124). This reviewer would like to confirm the authors whether the use of Complete Freund's Adjuvant and Incomplete Freund's Adjuvant were really permitted to apply for immunization. Complete Freund's is usually not used because it is a Category C or D animal experiment. How does the guideline of the authors’ institute state regarding the use of Complete Freund's Adjuvant?
  12. Line 184: Please explain the purpose of the use of a commercially available monoclonal antibody and what antibody was used.
  13. Line 205: Please specify the source of the antibody, like “rabbit antibody”.
  14. L1ne 209: The term “2nd antibody” had better be described as “enzyme labeled 2nd antibody.
  15. Lines 222-224, 262-264, and 268-270: One symbol “*” indicates the two items. Please use separately like *𝐵𝑚𝑒𝑎𝑛, and **σB or *𝐵𝑚𝑒𝑎𝑛, and #σB.
  16. Line 284: Please show “105” as “105” exponentially.
  17. Lines 294-298, and 299-305: Two animal experiments using Guiana pigs and swain sera are shown. Please consider moving these two parts from the validation section to animal study section.
  18. Line 299: Please show the number of field pig sera used for the validation.
  19. Lines 341-340, Figure 3: Please consider moving Figure 3 form the text to supplemental figures, because this has less discussing point.
  20. Lines 370-377, Figure 4: Please consider just showing the optimized conditions for the coating antigen concentration and the bocking BSA concentration here, and moving others from the text to supplemental figures.
  21. Lines 409-415, Figure 5: This reviewer did not understand the importance of Figure 5B using 5% skim milk, because the authors had already chosen the 5% BSA condition for ELISA. Please consider deleting Figure 5B.
  22. Lines 459-466, Figure 7: This reviewer did not understand the way how 14 red points in Figure 7B were obtained. Since the serum correlation between pig and guinea pig is weak, this reviewer suggests adding more results to generalize.
  23. Lines 490-495: Authors cite reference #47 concerning the competition ELISA, but reference #47 does not discuss about the PCV2 ELISA but the general discussion.
  24. Lines 581-582, Conflicts of Interest: The authors only declare that the research was conducted without any commercial or financial relationships, but do not declare that the authors belong to the WOOGENE Co., Ltd, which provides IMMUNIS® DMVac for ELISA antigen. Please consider describing this to have the transparency of this study.

Author Response

Major concern

The authors claim that the ELISA system specifically detects antisera against PDV2d because the antibody-antigen reactivity of ELISA is not influenced by PCV2a, PCV2b, or PRRSV antigens.

This reviewer thinks this experiment needs more clarification.

First of all, there is a high degree of identity and similarity among the capsid proteins of PCV2a, PCV2b, and PCV2d. Identity is 93% between PCV2d SNUVR201901 (MW623084) and PCV2a S9096 (JF317589), and 91% between PCV2d SNUVR201901 (MW623084) and PCV2b e320-1 (FJ905467). Similarity is 99% between PCV2d SNUVR201901 (MW623084) and PCV2a S9096 (JF317589), and 98% between PCV2d SNUVR201901 (MW623084) and PCV2b e320-1 (FJ905467). In these extremely high identity and similarity conditions, it is generally difficult to detect the type-specific antibody using the conventional hole capsid protein. The authors led the conclusion since no competition was found in the presence of PCV2a, PCV2b, and PRRSV antigens (Figure 5D-5F). However, this study lacks a competition experiment to assess the antigen-antibody reaction, as the homologous PCV2d antigen hampers ELISA.

  • We thank the reviewer for this critical and insightful comment. We fully agree that the high degree of sequence identity among PCV2 genotypes—specifically, >90% identity and >98% similarity between PCV2a/b and PCV2d capsid proteins—presents a valid concern regarding type-specific antibody discrimination using whole-virus immunogens.

In response, we clarify that the aim of the current study was to establish an in-house ELISA system for internal quality control purposes, not for genotype-specific diagnosis. The specificity validation in this study was performed using competitive ELISA against heterologous antigens (PCV2a, PCV2b, and PRRSV), and no measurable OD signal reduction was observed at either 1:1 or 1:5 competition ratios. These results support a lack of cross-reactivity under our validated assay conditions, demonstrating the assay’s analytical specificity within its intended application.

However, we acknowledge the reviewer’s point that homologous antigen competition (using PCV2d) is essential to confirm epitope-specific binding and to distinguish true antigen–antibody interactions from potential non-specific background, especially in the context of whole-virus immunogens. To address this, we have revised both the Results (Section 3.5) and Discussion to explicitly recognize this limitation and to state that homologous competition assays using PCV2d antigens will be incorporated in future studies to further validate target-binding specificity.

These clarifications are now included in the revised manuscript (Results Section 3.6, and 3.7 [Lines 445–471, Lines 472-508] and Discussion [Lines 563–583]), ensuring transparency about assay limitations while outlining our plan for experimental reinforcement.

Minor comments

  1. Lines 2-4, Title: Please specify the ELISA target as an anti-PCV2d antibody.
  • Thank you for your helpful suggestion. We agree that the original title did not clearly indicate the ELISA target. To improve clarity and specificity, we have revised the title to: (Line 2-4)

“Validation and Correlation Analysis of an In-House ELISA for Anti-PCV2d IgG Detection Using Biological Samples and Virus Culture Samples”

This revised title more accurately reflects the aim of the study, which is to detect anti-PCV2d antibodies using the developed in-house ELISA. We appreciate your comment, which helped enhance the precision of our manuscript.

  1. Lines 24-25, Abstract: As shown in the above Major concern, this reviewer thinks this experiment needs positive control to confirm the validity of the competition experiment.
  • We thank the reviewer for this important comment. We fully agree that the inclusion of a positive control using the homologous PCV2d antigen in competition ELISA is necessary to confirm the assay’s ability to detect antigen-specific interactions. In the current study, competitive ELISA was conducted using heterologous antigens (PCV2a, PCV2b, and PRRSV) to evaluate analytical specificity and cross-reactivity under validated in-house ELISA conditions. As described in the Results and Discussion, no measurable OD suppression was observed even at 1:1 and 1:5 competition ratios, supporting minimal cross-reactivity.

However, we recognize that the absence of a homologous competition condition (i.e., PCV2d antigen as a positive competitor) limits full verification of the assay’s epitope-level specificity. This concern has been addressed in the revised manuscript by adding the following clarification to both the Abstract and Results sections:

“Although the current competitive ELISA confirmed minimal cross-reactivity with heterologous antigens, additional competition assays using homologous PCV2d antigen are planned to further validate the specificity of antigen–antibody interaction.”

We have also added this point in the Discussion to outline our intention to incorporate homologous PCV2d competition in future studies as a positive control to more directly confirm target-specific blocking.

These additions acknowledge the limitation while clearly presenting our ongoing effort to enhance scientific rigor through further validation experiments.

  1. Lines 38-39: Please add the genus name for PCV, like the genus Circovirus family Circoviridae, together with the genus
  • Thank you for your helpful comment. We have revised the sentence to include the genus name as requested. The corrected sentence now reads: (Line 40-41)

“Porcine circovirus (PCV) is a non-enveloped, circular, single-stranded DNA virus, belonging to the genus Circovirus within the family Circoviridae.”

This addition ensures taxonomic accuracy and improves the clarity of the introduction. We appreciate your suggestion.

  1. Line 59: Over time, the predominant genotypes of PCV2 have slowly shifted from PCV2a to PCV2b to PCV2d, as shown in the text. However, views that PCV2d does not escape the immunity induced by PCV2a-based vaccines had better be added to the text [doi:10.1016/j.vetmic.2023.109796, doi.org/10.3390/vetsci6030061].
  • Thank you for your insightful comment. We fully agree that it is important to provide a balanced perspective regarding immune cross-protection between PCV2 genotypes. In response, we have revised the manuscript to include the recommended statement: (Line 63-66)

“Although PCV2d has become the predominant genotype in recent years, several studies report that commercial PCV2a-based vaccines can still provide partial or sufficient cross-protection against PCV2d, despite antigenic divergence.”

To maintain logical flow and avoid redundancy, we also reorganized the surrounding sentences to streamline the narrative and clarify that vaccine efficacy against PCV2d may vary depending on field conditions and strain heterogeneity. These changes improve the scientific objectivity and coherence of the background section. Thank you again for this constructive suggestion.

  1. Line 92: A species name “ hyopneumoniae” should be described as “Mycoplasma hyopneumoniae”.
  • Thank you for your comment. We agree with your suggestion, and the species name has been corrected to “Mycoplasma hyopneumoniae” in the revised manuscript to ensure taxonomic accuracy and consistency with scientific nomenclature (Line 102).
  1. Line 109: Please explain why the authors used real-time reverse transcription-polymerase chain reaction (RT-qPCR) for detecting DNA viruses such as PCV2. PCV2 needs no reverse transcription before DNA amplification.
  • Thank you for your important observation. We agree that PCV2, being a DNA virus, does not require reverse transcription for its detection. However, in this study, we utilized the Prime-Q PCV2/PRRSV Detection Kit (Genet Bio, Republic of Korea), a commercially available multiplex diagnostic kit designed to simultaneously detect both PCV2 (DNA) and PRRSV (RNA) targets within the same reaction protocol. The kit incorporates a unified RT-qPCR procedure to streamline testing and improve diagnostic efficiency.

We have clarified this point in the revised manuscript to avoid confusion. (Line 124-129)

  1. Lines 113 and 119: A term ORF2 had better be shown as ORF2
  • Thank you for your comment. In response, we have revised the formatting of the term "ORF2" in the relevant lines to italic (ORF2), in accordance with convention when referring to gene names. We appreciate your attention to detail. (Line 131, 138)
  1. Line 116: Please check if the author really used the DNA assembly software, Lasergene Seqman Pro (DNA Star, USA) for the small DNA virus like PCV2.
  • Thank you for your comment. While it is true that PCV2 has a relatively small circular genome (~1.7 kb), we utilized Lasergene SeqMan Pro (DNASTAR, USA) not only for assembly but also for accurate alignment, quality trimming, and error correction of Sanger sequencing reads. The use of this software helped minimize potential base-calling errors and ensured high-fidelity sequence confirmation. We have clarified this purpose in the revised manuscript to avoid any misunderstanding regarding the software’s role in our analysis. (Line 133-136)
  1. Line 131, Section 2.4: Twenty guinea pigs are described to be used here, but eighteen guinea pigs in 6 groups (4+3+3+3+3+2=18) were used in Figure 1B. Please check the number of guinea pigs.
  • Thank you for pointing out this discrepancy. You are correct—the total number of guinea pigs used in this experiment was 18, not 20. The error was due to a typographical mistake in the Methods section. We have corrected the number in the revised manuscript to reflect the actual group composition shown in Figure 1B. We appreciate your careful review. (Line 148)
  1. Line 137, Section 2.4.1: One 4-week-old female NZW rabbit is described to use here, but a 3-month-old NZW rabbit is shown to use in Figure 1A. Please check the age of the guinea pigs used for the study.
  • Thank you for your careful review. You are correct—the rabbit used in this study was 3 months old, not 4 weeks as initially described. This was an oversight in the manuscript. We have corrected the age of the rabbit to "3-month-old" in Section 2.4.1 to ensure consistency with the data presented in Figure 1A. We appreciate your attention to detail. (Section 2.4.1, Line 155)

  1. Line3 138-142: It is described that “The animal experiment protocol was approved by the WOOGENE B&G Co., Ltd Institutional Animal Care and Use Committee (WG-IACUC-2024-004) and performed following the guidelines and regulations detailed by the committee.” (lines 122-124). This reviewer would like to confirm the authors whether the use of Complete Freund's Adjuvant and Incomplete Freund's Adjuvant were really permitted to apply for immunization. Complete Freund's is usually not used because it is a Category C or D animal experiment. How does the guideline of the authors’ institute state regarding the use of Complete Freund's Adjuvant?
  • Thank you for this important ethical inquiry. We confirm that the use of Complete Freund’s Adjuvant (CFA) and Incomplete Freund’s Adjuvant (IFA) was applied only in rabbit immunization for the generation of polyclonal antibodies. This protocol was thoroughly reviewed and approved by the Institutional Animal Care and Use Committee (IACUC) of WooGene B&G Co., Ltd (Approval No. WG-IACUC-2024-004), as detailed in the approved protocol and official IACUC approval documents (Supplementary Files: WGD-AE05-RC01 and WGD-AE05-RC03).

As stated in the protocol (pages 2–3), a single immunization using CFA was administered to induce a robust primary immune response, followed by boosters using IFA. This immunization procedure was explicitly categorized as a Category D experiment, in accordance with Korean national ethical standards and the institutional guidelines. Additionally, Xylazine was used for sedation and pain management, and veterinary staff continuously monitored animal well-being throughout the study.

We hope this clarifies the institutional and ethical compliance regarding CFA use in rabbits. (Line 160-165)

  1. Line 184: Please explain the purpose of the use of a commercially available monoclonal antibody and what antibody was used.
  • Thank you for your thoughtful and detailed comment. In this study, we utilized a commercially available monoclonal antibody, clone 11E59 (CAT. No. 9056, Median Diagnostics, Republic of Korea), which is an IgG2a, G3 isotype antibody generated in Balb/c mice and specifically targets the capsid protein of Porcine Circovirus type 2 (PCV2).

According to the manufacturer's specification, 11E59 reacts with multiple PCV2 genotypes including PCV2a, PCV2b, and PCV2d, but shows no cross-reactivity to PRRSV, and is suitable for both FA and ELISA applications. It was purified to ≥95% purity by Protein G sepharose chromatography.

In our study, the primary purpose of using 11E59 was to functionally verify the specificity and validity of the in-house polyclonal antibodies generated in rabbit and guinea pig after antigen purification and concentration. This monoclonal antibody served as a reliable standard reference to demonstrate that the OD signals obtained using in-house antibodies were due to specific interactions with PCV2d antigen rather than non-specific binding or procedural artifacts.

We have clarified this purpose and included detailed antibody information in Section 2.6 (Line 221-223) of the revised manuscript. We sincerely appreciate your careful review and the opportunity to strengthen the methodological rigor of our study.

  1. Line 205: Please specify the source of the antibody, like “rabbit antibody”.
  • Thank you for your comment. In response, we have revised the sentence to specify the antibody source as requested. The updated sentence now reads: (Line 225)

“The in-house antibody (WG-PCV2d pAb, Rabbit Anti-PCV2d IgG) was subjected to 10-fold serial dilutions, ranging from 1 mg/mL to 1 ng/mL.”

This revision improves clarity by explicitly indicating the species origin of the antibody. We appreciate your helpful suggestion.

  1. L1ne 209: The term “2nd antibody” had better be described as “enzyme labeled 2nd antibody.
  • Thank you for your suggestion. We agree that the term can be made more precise. Accordingly, we have revised the phrase to “enzyme-labeled 2nd antibody” in the manuscript to accurately reflect its function in the ELISA protocol. We appreciate your attention to clarity and detail. (Line 250)
  1. Lines 222-224, 262-264, and 268-270: One symbol “*” indicates the two items. Please use separately like *?????, and **σB or *?????, and #σB.
  • Thank you for your helpful suggestion. To improve clarity and avoid confusion, we have revised the notation by using separate superscript symbols for each term. Specifically, we now use *Bmean to indicate the mean of the blank OD, #σB to indicate the standard deviation of the blank, *σintra to indicate the Standard deviation of replicate OD values within a single plate, and #μintra to indicate the Overall mean OD value across plates. These symbols have been consistently applied throughout the text, including in the LOD formula and corresponding figure legend. We appreciate your attention to detail. (Line 262-264, 288-290, 303-305, and 308-310)
  1. Line 284: Please show “105” as “105” exponentially.
  • Thank you for your suggestion. We have corrected the notation and now present “10⁵” in exponential format in the revised manuscript as requested. (Line 324)
  1. Lines 294-298, and 299-305: Two animal experiments using Guiana pigs and swain sera are shown. Please consider moving these two parts from the validation section to animal study section.
  • Thank you for your valuable suggestion. In response, we have added a new subsection (Section 2.4.3 (Line 187-199)) under the Animal Study section to describe the use of guinea pig and field pig sera for ELISA validation. This change clarifies the origin and purpose of the biological samples and improves the structural coherence of the Methods section. The Validation section now focuses solely on the analytical performance of the ELISA system. We appreciate your recommendation.
  1. Line 299: Please show the number of field pig sera used for the validation.
  • Thank you for your comment. We have now specified the number of field pig sera used for validation. A total of 30 serum samples (n = 10 per year) collected from 2022 to 2024 were tested using the in-house ELISA and compared to commercial kit classifications. (Line 193-194)
  1. Lines 341-350, Figure 3: Please consider moving Figure 3 from the text to supplemental figures, because this has less discussing point.
  • Thank you for your suggestion. We agree that Figure 3 primarily serves as a technical validation of the antibody purification process rather than a core result for scientific discussion. In response, we have moved Figure 3 to the supplementary materials (now Supplementary Figure S1 (Line 623-632) to improve the focus and flow of the main text. The corresponding figure legend and references in the Results section have been updated accordingly.
  1. Lines 370-377, Figure 4: Please consider just showing the optimized conditions for the coating antigen concentration and the blocking BSA concentration here, and moving others from the text to supplemental figures.
  • We appreciate the reviewer’s insightful suggestion. In the revised manuscript, we have followed your recommendation by reorganizing the data presentation for improved clarity and focus.

Specifically, Figure 3A now presents the ROC curve generated under the optimized ELISA conditions (10⁵ FAID₅₀/mL coating antigen, 5-minute substrate incubation, and BSA blocking). Figure 3B presents a comparative ROC analysis across four BSA concentrations under the same standardized coating and incubation conditions, illustrating the impact of blocking composition on diagnostic accuracy.

To maintain the scientific integrity of the full optimization process while avoiding excessive complexity in the main text, we have moved the remaining experimental conditions—i.e., full ROC comparisons across coating antigen concentrations (10⁴–10⁶ FAID₅₀/mL), substrate incubation times (3–10 min), and BSA concentrations—to Supplementary Figure 2 (S2), and skim milk concentrations to Supplementary Figure 3 (S3) with a detailed caption (Line 633-640, Line 641-649).

These changes allow Figure 3 to highlight only the most diagnostically relevant parameters while preserving transparency and reproducibility through supplementary materials. We trust this restructuring aligns with your recommendation.

  1. Lines 409-415, Figure 5: This reviewer did not understand the importance of Figure 5B using 5% skim milk, because the authors had already chosen the 5% BSA condition for ELISA. Please consider deleting Figure 5B.
  • We appreciate the reviewer’s observation. Figure 5B was originally included to provide a scientific comparison between 5% skim milk and 5% BSA—both widely used blocking agents—to support the selection of BSA as the optimal blocking condition for this in-house ELISA.

However, we agree with the reviewer that this figure may be redundant, given that 1–5% BSA had already been thoroughly optimized in earlier experiments (see Figure 3 and Supplementary Figure [S2-S3]). To improve clarity and conciseness of the main figures, we have removed Figure 5B from the revised manuscript and made corresponding adjustments to the Results section and figure enumeration.

We thank the reviewer for this helpful suggestion. (Line 438-444)

  1. Lines 459-466, Figure 7: This reviewer did not understand the way how 14 red points in Figure 7B were obtained. Since the serum correlation between pig and guinea pig is weak, this reviewer suggests adding more results to generalize.
  • We appreciate the reviewer’s comment and the opportunity to clarify Figure 7B (Revised 6B). The 14 red points shown in the figure correspond to matched serum pairs collected from guinea pigs and pigs that were either vaccinated with the same antigen lot or confirmed to be positive for PCV2d antibodies. Guinea pig sera were generated under controlled immunization using fractional doses of a validated inactivated vaccine, whereas porcine sera were selected from a repository of field samples (2022–2024) that had been pre-classified using a commercial ELISA.

The intention of this correlation analysis was not to imply strict quantitative equivalence across species, but rather to demonstrate cross-species predictive validity under controlled immunization and validated assay conditions. While the R² value (0.1815) indicates a moderate correlation, we further included a Bland–Altman analysis (Figure 6C) to assess the degree of agreement between the two datasets. This method is widely recognized for validating diagnostic tools with different measurement scales [1,2], and showed acceptable agreement with a mean bias of −0.21 and 95% limits of agreement from −0.48 to 0.07.

Importantly, the primary purpose of this dataset is not to generalize across all field conditions but to support the feasibility of using guinea pig antisera for internal lot-release verification during vaccine manufacturing. To clarify these points, we have revised the figure panel (now Figure 6), updated the legend, and modified the Results section to explain the sample selection and analytical intent more explicitly (Line 472-507).

  1. Lines 490-495: Authors cite reference #47 concerning the competition ELISA, but reference #47 does not discuss about the PCV2 ELISA but the general discussion.
  • We thank the reviewer for identifying this discrepancy. We acknowledge that the previously cited reference (#47: Plested et al., 2003) pertains to general ELISA protocols and does not specifically address PCV2-related ELISA optimization. To address this, we have replaced the citation with a more directly relevant and recent study by Liu et al. (2025), which reports the development and optimization of a PCV2d-specific blocking ELISA. In their study, bovine serum albumin (BSA) was identified as the optimal blocking agent, showing lower background OD values and improved assay reproducibility compared to skim milk. This finding is in line with our own results and thus provides stronger scientific support for our methodology.

The revised sentence now reads: (Line 528-531)

“Blocking agents were systematically compared, and BSA consistently resulted in lower background OD values, improved signal-to-noise ratios, and superior intra-/inter-assay reproducibility compared to skim milk, consistent with previous PCV2d-specific immunoassay optimization studies [Liu et al., 2025].”

We have updated the reference list accordingly.

  1. Lines 581-582, Conflicts of Interest: The authors only declare that the research was conducted without any commercial or financial relationships, but do not declare that the authors belong to the WOOGENE Co., Ltd, which provides IMMUNIS® DMVac for ELISA antigen. Please consider describing this to have the transparency of this study.
  • Thank you for your valuable comment. We agree that it is important to ensure full transparency regarding institutional affiliations and material sources. Accordingly, we have revised the Conflicts of Interest statement as follows: (Line 668-672) “The authors are employees of WOOGENE B&G Co., Ltd., which supplied the IMMUNIS® DMVac used as the ELISA antigen in this study. However, the research was conducted independently without any commercial or financial relationships that could be construed as a potential conflict of interest.” We appreciate your suggestion, which has helped improve the transparency and clarity of our disclosure.

Round 2

Reviewer 1 Report

Comments and Suggestions for Authors

The manuscript has been improved. For the benefit of the reader however, a number of points need clarifying and certain statements require further justification. These are given below.

  1. The title should be more improved, more concisely but it should clarify PCV2d. No typical reader recognizes PCV2d.
  2. Line 12. PCVAD should be spelled out.
  3. Lines 18-19. No typical reader recognizes MMUNIS® DMVac.
  4. Line 23. Genomic copies should be better than FAID₅₀, if they are matched. The authors never indicated the relationship between them.
  5. Lines 65-66. This sentence should be modified.
  6. Lines 101-102. Severe fever thrombotcytopenia syndrome virus.
  7. Line 103. TCID₅₀ and FFU should be spelled out.
Comments on the Quality of English Language

The language is difficult to understand and still requires editing for clarity. Correction of the paper by a native English speaker is recommended.

Author Response

The manuscript has been improved. For the benefit of the reader however, a number of points need clarifying and certain statements require further justification. These are given below.

1. The title should be more improved, more concisely but it should clarify PCV2d. No typical reader recognizes PCV2d.

  • Thank you for your helpful suggestion. We have revised the title to improve clarity and conciseness, and to provide the full name of PCV2d upon first mention. The revised title now reads:

“An In-House ELISA for Anti Porcine Circovirus Type 2d (PCV2d) IgG: Analytical Validation and Serological Correlation”

This change improves readability for a broader audience while maintaining scientific accuracy.

2. Line 12. PCVAD should be spelled out.

  • Thank you for pointing this out. We have revised the sentence to spell out the abbreviation upon its first mention. The term now reads as “porcine circovirus-associated disease (PCVAD)” in (Line 13) the revised manuscript.

3. Lines 18-19. No typical reader recognizes MMUNIS® DMVac.

  • Thank you for the comment. To enhance clarity, we have revised the sentence to include a brief explanation of IMMUNIS® DMVac upon its first mention. The updated sentence now reads: (Line 20)

“Guinea pigs (n = 18) were immunized with IMMUNIS® DMVac, an inactivated PCV2d vaccine candidate developed by WOOGENE B&G, at different doses.”

This revision helps clarify the nature and origin of the vaccine for readers unfamiliar with the product.

4. Line 23. Genomic copies should be better than FAID50, if they are matched. The authors never indicated the relationship between them.

  • Thank you for your valuable comment. We agree that quantifying the viral input using genomic copies provides a more standardized and reproducible metric. Accordingly, we have revised the manuscript to express the viral inoculum in terms of genomic copies, which were quantified by qPCR. In addition, we have clarified in the Methods section that the genomic copy number was determined from the same PCV2d virus stock used for calculating FAID50, allowing both values to be correlated. The revised sentence now reads: (Line 24)

“The PCV2d antigen used for immunization contained 1 × 105 genomic copies/mL, quantified by qPCR from the same virus stock used for determining the FAID50.”

5. Lines 65-66. This sentence should be modified.

  • Thank you for your suggestion. We agree that the original sentence lacked clarity regarding the interpretation of neutralization assay results. We have revised the sentence to more precisely reflect the implication of antigenic variability among PCV2d field strains. The updated sentence reads as follows: (Line 69-72)

“Moreover, neutralization assays have demonstrated that certain field strains of PCV2d exhibit reduced neutralizing reactivity to antibodies induced by PCV2a-based vaccines, indicating potential antigenic variability within the PCV2d genotype [28].”

6. Lines 101-102. Severe fever thrombotcytopenia syndrome virus.

  • Thank you for pointing this out. We have corrected the term by adding “virus” to ensure proper nomenclature. The revised phrase now reads: “severe fever with thrombocytopenia syndrome virus” in (Line 108) of the revised manuscript.

7. Line 103. TCID₅₀ and FFU should be spelled out.

  • Thank you for your comment. We have revised the sentence to spell out the abbreviations “TCID50” and “FFU” as “tissue culture infectious dose 50” and “focus-forming units,” respectively. We also clarified the role of FAID50 as a semi-quantitative indicator based on immunofluorescence-based detection of cytopathic effects. The revised sentence now reads: (Line 108-112)

“Although fluorescent antibody infectious dose 50 (FAID50) is not directly equivalent to tissue culture infectious dose 50 (TCID50) or focus-forming units (FFU), it serves as a semi-quantitative indicator of infectious viral concentration, based on the cytopathic effect detected by immunofluorescence staining.”

General Note Regarding Language Editing:
We acknowledge the reviewer’s suggestion to improve the clarity of the English language. In response, we plan to request professional English editing through MDPI’s Author Services before final acceptance of the manuscript. The revised version will reflect the necessary improvements in grammar and style to ensure that the research is clearly and accurately communicated.

Reviewer 2 Report

Comments and Suggestions for Authors

The authors well responded this reviewer's comments and suggestions and revised the manuscript.

Now, this reviewer would like to  recommend the  accept of this manuscript.

Author Response

Comment:
The authors well responded to this reviewer’s comments and suggestions and revised the manuscript.
Now, this reviewer would like to recommend the accept of this manuscript.

Response:
We sincerely thank the reviewer for their thoughtful evaluation and kind recommendation for acceptance. We greatly appreciate the time and expertise provided during the review process, which helped us to significantly improve the clarity and scientific rigor of our manuscript.